# Against All Odds: How Eindhoven Emerged as a Deeptech Ecosystem

A. Georges L. Romme

Industrial Engineering & Innovation Sciences Department, Eindhoven University of Technology, P.O. Box 513, 5600 MB Eindhoven, The Netherlands; a.g.l.romme@tue.nl; Tel.: +31-40-247-2170

**Abstract:** The Brainport-Eindhoven region has developed into a leading location for deeptech entrepreneurship in Europe. Against all odds, it has transformed itself from a region that heavily depended on the multinational company Philips, into a diverse and fast-growing deeptech ecosystem. While this success has not gone unnoticed, there is not yet a clear account of how and why the Eindhoven region emerged as a global hotspot for deeptech innovation and entrepreneurship. Moreover, such an account might provide an exemplary model of a collaborative ecosystem, one that provides an alternative to the "winner-takes-all" entrepreneurial culture of Silicon Valley. This essay explores the performance of the Eindhoven region in terms of three structural conditions. First, the focus on deeptech R&D and entrepreneurship appears to be deeply rooted in the region's history as well as strong competencies in systems engineering, design thinking, and multidisciplinary collaboration. Second, a collaborative approach to regional policy gives industrial, academic, and governmental actors an equivalent position in its "triple helix" governance. Finally, the Eindhoven region benefits from a systemic approach toward co-locating R&D and entrepreneurial activities on five campuses. Overall, the huge complexity of deeptech systems and products apparently requires a truly collaborative approach at all levels of the entrepreneurial ecosystem.

**Keywords:** deeptech; systems thinking; entrepreneurial ecosystem; valley of death; co-location; digitalization; Silicon Valley; systems engineering; smart specialization

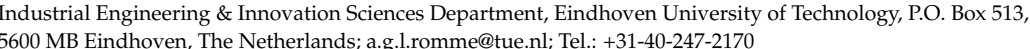

*"The next Silicon Valley could well be in Eindhoven"* [1]

Eindhoven is *"hands down the most inventive city in the world"* [2]

## 1. Introduction

The Brainport-Eindhoven region accounts for more than 25 percent of all R&D investments by industrial firms in The Netherlands [3], with a regional population that amounts to less than 5 percent of the total Dutch population. Moreover, the Brainport-Eindhoven region has the highest number of patents granted per inhabitant in the world [1,4]. This region is especially known for its *deeptech entrepreneurship*, which draws on breakthrough technologies in new synthetic materials, artificial intelligence, embedded software, mechatronics, precision engineering, and other disciplines [5,6].

The "next nature" perspective [7] serves to frame and understand the rise of deeptech technology and its pervasive societal impact. This perspective transforms the traditional idea of nature (one traditionally assumed to be opposed to technology), based on the observation that nature/biology and technology are increasingly merging, or even trading places. Technology therefore can be conceived as a "next nature" that is often taken for granted [7]. An example is how the Internet of Things (IoT) already prevails in our cars, offices, and homes: one's car, for example, directly communicating a technical deficiency to the garage, Netflix sending suggestions for what to view tonight, and so forth—in ways that we often take for granted. In a metaphorical sense, these extremely complex and

layered systems of hardware and software increasingly operate below the surface of human attention, which is another reason to call them deeptech.

While the emergence of Brainport-Eindhoven as a deeptech ecosystem has not gone unnoticed [2,8–10], a clear account of how and why this region arose as a globally leading hotspot is not yet available. Such a descriptive account would be valuable because the Eindhoven region's setting and history appear to be fundamentally different from those of Silicon Valley, the global benchmark for developing technology-driven ecosystems [11–13]. As such, the Eindhoven region has the potential to provide the archetype of a collaborative ecosystem, one that may provide an alternative to the "winner-takes-all" entrepreneurial culture of Silicon Valley [14,15]. This alternative is especially important in the continental European setting, which does not have a platform economy largely based on software innovations, but rather a manufacturing economy that also draws on hardware innovations.

The next section outlines the main challenges arising from pursuing a deeptech regional profile, especially drawing on the literature about entrepreneurial ecosystems and the so-called "valley of death" in deeptech entrepreneurship. Subsequently, the Eindhoven ecosystem is described and assessed in terms of its historical conditions, the specific competencies required for deeptech entrepreneurship, the location-boundness of deeptech activities, and so forth. In addition, several future challenges for the Eindhoven ecosystem are explored. Finally, I assess whether the Eindhoven region provides an alternative to the global benchmark of Silicon Valley. In the remainder of this essay, "Eindhoven" is used to refer to the broader Brainport region that includes 21 municipalities, of which Eindhoven is the central and largest one.

## 2. Background: Main Challenges in DeepTech Entrepreneurship

This section serves to outline the theoretical background of the argument in this essay. From the perspective of systems theory, two discourses appear to be relevant: the smart specialization literature and the entrepreneurial ecosystem literature, both discussed in the first subsection. Subsequently, Section 2.2 draws on the "valley of death" framework developed in the entrepreneurship literature to better understand the main (ecosystem) challenges arising from deeptech entrepreneurship.

### 2.1. Smart Specialization and Entrepreneurial Ecosystems

Regional profiles and policies in innovation and entrepreneurship have, especially in Europe, been increasingly informed by the so-called "smart specialization" perspective [16,17]. *Smart specialization* involves the capacity of an economic system (e.g., a region) to generate new specialties "through the discovery of new domains of opportunity and the local concentration and agglomeration of resources and competencies in these domains" [18] (p. 1). In this respect, the rationale of smart specialization is to build upon existing structures and transform them by means of new (but related) R&D activities [18]. Smart specialization theory thus suggests that a regional system cannot be created from scratch, but, rather, needs to exploit extant structures and resources in any major transformation to a new system state.

Another relevant perspective arises from the entrepreneurial ecosystem literature. Stam defines an *entrepreneurial ecosystem* as "a set of interdependent actors and factors coordinated in such a way that they enable productive entrepreneurship" [19] (p. 1765); he also presents a conceptual framework of various conditions as well as outputs and outcomes of an entrepreneurial ecosystem. Subsequent work has served to further develop and validate the entrepreneurial ecosystem construct [20–22]. Moreover, Stam et al. [23] conducted an extensive comparative study of five Dutch entrepreneurial ecosystems (including Eindhoven), observing that the Eindhoven ecosystem increased its performance remarkably since the global financial crisis in 2008. For example, economic data for the period 2004–2014 demonstrate that the Eindhoven region increased its average added value (per fte) by more than 25 percent in the years after 2008. The Eindhoven ecosystem also tops all other Dutch ecosystems in terms of project intensity, as measured by the number of

innovation projects per thousand companies [23]. Data on the knowledge structure suggest the Eindhoven ecosystem is dominated by three large high-tech companies (NXP, Philips, and ASML) and two higher education institutes (TU/e and Fontys) [23].

### 2.2. Valley of Death in DeepTech Entrepreneurship

The entrepreneurship literature provides another helpful perspective, in the form of the *valley of death* framework [24]. This framework, outlined in Figure 1, helps one to understand the major challenges and barriers that deeptech startups and companies face. In this respect, deeptech startups are ventures characterized by extremely "high risks" and (potentially) "high benefits": they constitute a huge risk for the entrepreneurs and investors involved, but they can result in immense benefits for society at large, as well as for the people driving and owning the venture.

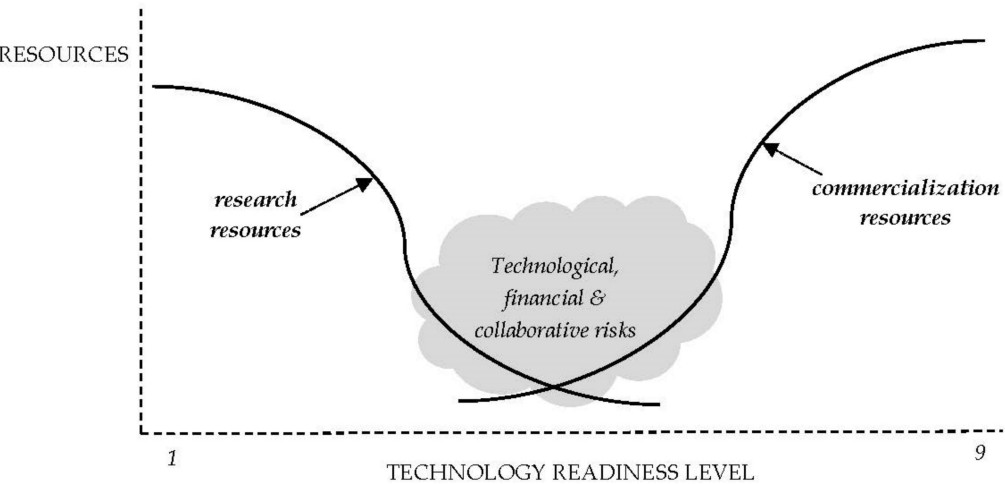

**Figure 1.** The Valley of Death in Deeptech Entrepreneurship.

The high risk of deeptech venturing is evident from its high failure rates. Whereas specific data for deeptech ventures are not available, previous studies have collected data on the broader population of technology-driven ventures. For example, Marmer et al. [25] observed that more than 90 percent of these ventures fail to enter the market, whereas Song et al. [26] reported a failure rate of around 80 percent. It is very likely that deeptech ventures are the most risky ones in this population, and therefore have failure rates above 90 percent. As such, deeptech venturing is extremely challenging, because these ventures:

- develop products and systems with a *very high technological complexity*, arising from the combination of extremely complex hardware and software [5];
- have a long time-to-market of usually at least 3 to 5 years—but often much longer—and thus require *major investments*, in terms of both financial and human resources: in terms of financial resources, a typical deeptech venture needs EUR 10M to 20M in the first (series A) investment round, and hundreds of millions in the second investment round [27];
- often require extensive *innovation ecosystems* (or clusters of collaborating firms) around the deeptech value proposition of the focal venture, in which various suppliers also have to invest in developing new components and services [28–30].

Each of these three challenges constitutes a potential cause for failure of a deeptech venture: (a) the *technological* risk of failure, arising from the extremely high complexity of the product/system being developed; (b) the *financial* risk of not acquiring the investment volume required; and (c) the *collaborative* risk of not being able to obtain the buy-in of all suppliers and other co-creating stakeholders needed. These three risks are, of course, somewhat interdependent: for example, if the prototyped technology does not perform adequately, investors are not likely to come on board, and a key collaborator that cannot

supply a critical component increases the technological risk, and so forth. In any case, deeptech ventures are characterized by very high-risk levels on all three dimensions.

The three challenges and associated risks converge in the "valley of death" of deeptech entrepreneurship, as visualized in Figure 1. The horizontal axis in Figure 1 represents the time-to-market, operationalized in terms of Technology Readiness Level (TRL). The vertical axis reflects the resources needed to develop the initial technology, and, at a later stage, the resources the venture needs to apply in order to commercialize this technology. Many deeptech ventures do not survive the gap between the initial (e.g., academic and/or pre-seed funding) resources and the resources (possibly) obtained later for commercializing the technology [24,31]. In terms of the TRL construct, the valley of death typically starts at TRL 4 and ends around TRL 7 [32]. The "valley of death" framework thus provides a helpful perspective on the major challenges that deeptech ecosystems are exposed to.

## 3. Materials and Methods

In this essay, I draw on three sources of data. First, earlier work includes studies of the largest and most prominent campus in the Eindhoven region [33,34] and a study for the Organisation for Economic Co-operation and Development (OECD) in which five Dutch entrepreneurial ecosystems (incl. Eindhoven) were analyzed and compared [22,23]. The latter study draws on both quantitative and qualitative data regarding regional governance, knowledge flows, collaborative practices, and other key aspects of these five ecosystems.

Second, this essay also draws on more than twenty MSc thesis projects on various innovation and entrepreneurial practices in the Eindhoven ecosystem, all (co)supervised by the author of this article. For the findings reported in the remainder of this paper, the ten most important MSc theses are [35–44]. These thesis projects employed case study methods, mathematical simulation models, surveys, and other research methods to codify and analyze various entrepreneurial practices. For more detailed information, I refer to the full thesis reports [35–44], all available in open access.

Third, this essay benefits from more than 15 years of participant-observation [45] by the author. For example, preliminary versions of the assessment described in the next section were used in many presentations and discussions with visiting delegations from other regions (e.g., [46]), which allowed the author to further validate the findings and conclusions about the Eindhoven ecosystem reported in this paper. Notably, the evidence collected in the first two sets of studies forms the backbone of the key argument in the next section; the participant-observation role primarily served to triangulate this key argument by obtaining input from outsiders (e.g., with extensive knowledge of how other regional ecosystems operate).

## 4. Main Findings

In this section, the Eindhoven ecosystem is described and assessed in terms of its historical and economic context, key competencies, regional governance, location-based innovation activity, and selectivity.

### 4.1. Historical and Economic Context

The Eindhoven region, also known as the South-East Brabant, was long characterized by harsh living conditions, because its sandy soils were not suitable for agriculture. These poor conditions formed the basis for the emergence of various cooperative organizations, in which peasants and other inhabitants sought to collectively address the harsh living conditions. In the second half of the 19th century, the region became more attractive for industrial companies, due to the construction of a canal and various roads, in combination with a large labor supply and rather low wages [23]. The first generation of industrial companies mainly involved manufacturers of textiles, cigars, and matches [47]. In 1892, Philips started building a small factory for light bulbs, which subsequently evolved into a multinational company. In the absence of adequate infrastructure in the region, Philips invested in building completely new neighborhoods, schools, and other social facilities, in

order to be able to attract thousands of employees (with their families) from other Dutch and European regions [48]. Other larger companies in the region, such as VDL, DAF (Trucks) and Brabantia, also benefited from the growing attractiveness of the region. After the Second World War, Philips started pressing the Dutch government to establish an institute for the training of engineers, which resulted in a new college being founded in 1956; this college later became Eindhoven University of Technology (TU/e).

In the 1980s, Philips divested its lithographic technology (developed in its R&D unit) by founding ASML, which subsequently developed into the world's largest producer of lithographic machines for the semiconductor industry. At the turn of the century, Philips also divested its manufacturing activities in semiconductors, optics, and X-ray systems; these divestments formed the basis for several new deeptech companies such as NXP, FEI, and Malvern Panalytical. In the past thirty years, the knowledge infrastructure of the region also became more diverse when institutes such as the Holst Centre, Netherlands Organization for Applied Scientific Research (TNO), and Dutch Institute for Fundamental Energy Research (DIFFER) were founded in and/or relocated to the Eindhoven region; moreover, Fontys Hogescholen grew into a major institute for higher education, one that also offers a large number of engineering programs.

Overall, the historical and economic context of the region appears to provide fertile conditions for collaborative arrangements around (deeptech) innovation and entrepreneurship. The region's unique capabilities in deeptech entrepreneurship are also increasingly acknowledged by leading research institutes like the nuclear research organization CERN and the European Space Agency. For example, HighTechXL, a deeptech venture builder located in Eindhoven, has been creating various new ventures based on breakthrough technologies developed at these two institutes. Another example is VDL's venture in proton therapy, based on CERN's core technology, which serves to treat tumors in the most difficult-to-reach (e.g., the brain) locations in the human body.

Today, the so-called Brainport region (or South-East Brabant) contains around 785,000 inhabitants living in 21 municipalities, of which Eindhoven (with 238,000 inhabitants), Helmond (93,000), and Veldhoven (46,000) are the three largest ones.

### 4.2. Key Competencies Required for DeepTech Entrepreneurship

While the Eindhoven region still is often associated with the former consumer products of Philips, today it mainly involves companies like ASML, NXP, VDL, Signify, and Prodrive Technologies, all of which develop advanced technical products and systems for business-to-business (B2B) industrial markets, characterized by high complexity and low volumes. Eindhoven's entrepreneurial profile on this type of market is one of *deeptech entrepreneurship* [5], as outlined in Section 1. Deeptech companies combine multiple technologies in new solutions in, for example, healthcare, energy storage, robotics, or internet of things. As such, deeptech products/systems use unique and well-protected innovations in, for instance, new synthetic materials, artificial intelligence, embedded software, mechatronics, electronics, photonics, and precision engineering [5,6].

The most prominent example of deeptech technology in the region is ASML's lithographic machine, which uses light to print tiny patterns on silicon; this is an extremely critical step in mass-producing microchips, one in which very complex hardware meets complex software. ASML has been making giant leaps on this tiny scale in the past few decades, in ways that competitors have not been able to match. As a result, ASML today dominates the entire semiconductor industry as the main supplier of lithographic machines (to, e.g., Intel, Samsung, and TSMC). Recently, ASML also became a key issue in international politics when the U.S. government became aware of ASML's critical position in the value chain of all semiconductor manufacturers in the U.S. [49], which resulted in a Dutch ban, demanded by the U.S. government, on the export of ASML's lithographic machines to China [50]. Other examples of deeptech entrepreneurship in the Eindhoven region are Lightyear's solar-powered electric vehicles that are built for grid independence and NXP's

near-field (i.e., wireless proximity) technology that is integrated in billions of smartphones, tablets, and other electronic products and systems.

The development of deeptech solutions for highly complex challenges in industrial markets requires strong competencies in *systems engineering* and *design thinking*. Systems engineering and design thinking are crucial for creative processes in which many different disciplines work together and technologies are combined [51,52]. Systems engineering and design thinking together provide an interdisciplinary approach that focuses on how to design, integrate, and manage a complex system over its entire life cycle, using a large number of tools, including modeling and simulation, architecture development, requirements analysis, quality assurance, and scheduling to manage complexity [53,54]. Whereas systems engineering is taught in many universities of technology, advanced professional skills in this area can only be acquired by engaging in this type of work, often across several decades.

Strong competencies in systems engineering and design thinking may often be difficult to develop because they also require well-developed skills and networks in *multidisciplinary collaboration* between industry and knowledge institutions. Multidisciplinary collaboration is widespread in industry, simply because most products require input and expertise from a large number of specialized disciplines. However, in academia this type of work often suffers from a lack of appreciation for the contributions of colleagues from other disciplines [55]. In the Eindhoven ecosystem, competencies in systems engineering, design thinking and multidisciplinary collaboration are well-developed, with many successful examples and role models that sustain its deeptech innovation culture [56,57].

### 4.3. Regional Collaboration and Governance

As described in 4.1, in the 1980s Philips started divesting from a substantial number of activities. Around the turn of the century, these divestments and their implications for regional employment led to a collective search for how Eindhoven and 20 adjacent municipalities could become less vulnerable to economic fluctuations and also less dependent on a single multinational company [58]. This joint effort by local governments, industrial companies and knowledge institutions initially resulted in the so-called Stimulus program and later the Horizon program [59]. The latter provided the basis for the Brainport foundation, established in 2006. The Stimulus and Horizon programs involved (recommendations regarding) the transformation of Philips' NatLab campus into an open campus, new concepts and initiatives for living and working (incl. the new city quarter Brandevoort and Flight Forum business park), and clustering and improving cooperation between suppliers and buyers (outsourcers) on the various deeptech markets that the region focuses on. Over the past twenty years, the vast majority of these ideas and initiatives were implemented and realized. Brainport-Eindhoven has thus grown into an appealing example of triple helix collaboration between local government, industry, and knowledge institutions [58,60,61].

This collaboration in regional policy development has been implemented in a professional and stable manner in the Brainport (Development) foundations [58]. Representatives of the business community, knowledge institutions, and municipalities participate as equivalent partners in the governance of these Brainport agencies. Moreover, Brainport has its own operational organization [62], which appears to be rather unique because, in other regions, the triple helix collaboration is typically arranged as an extension of the largest municipality, with operational support from the civil servants of this municipality [23]. This is highly undesirable if local governments, industry, and knowledge institutions want to collaborate on an equal footing, as in the Eindhoven case [60].

### 4.4. Campus Locations as Physical Hotspots for DeepTech Innovation

In addition to its approach to regional governance, the Eindhoven region also benefits from substantial innovation capabilities at multiple physical locations (or hotspots) for innovation and entrepreneurship. The most important hotspots are the TU/e campus, Automotive Campus, Strijp-S, Brainport Industries Campus, and High Tech Campus Eind-

hoven (HTCE). These location-specific hotspots promote the transfer of tacit knowledge [63] and, moreover, the co-location of R&D is a critical condition for effective (e.g., industrial-academic) collaboration between different disciplines [64]. Each of the five campuses has its own signature and functionality in the field of deeptech knowledge development and entrepreneurship.

The criticality of co-locating various activities is best illustrated in terms of the design of the HTCE [33], the largest and most prominent campus in the region. For one, the HTCE was created around a social hub, called "The Strip", which includes many restaurants, shops, conference facilities, and additional services. All lease agreements with residents stipulate that catering facilities cannot be offered in the offices and labs in buildings elsewhere on the campus; as a result, many thousands of knowledge workers walk to The Strip every day for lunch or other (social or business) activities. This promotes informal conversations and knowledge exchanges between (employees of) companies and institutes on the HTCE [33,64] because many people are stimulated to frequently move out of their direct work environment, develop new contacts, and so forth. In addition, the HTCE has collective facilities such as cleanrooms and test instruments that can be rented per day. This turns the campus into an attractive location for deeptech startups that need these facilities for various reasons (e.g., to test their prototypes) but that cannot yet afford to purchase and build such facilities themselves. Finally, the HTCE hosts a large number of knowledge brokers, such as EIT Digital, ARTEMIS, ITEA, and HighTech NL. These intermediaries deliberately seek to connect and bring together various parties that may be able to develop entirely new value chains [33]. Some examples of new value chains arising from this campus are Solliance (regarding a new thin-film generation of solar cells/panels), various startups working on specific applications of photonics [65], and LifeSense (developing health solutions in the form of smart textiles). Some intermediaries, for example, the Embedded Systems Institute (ESI) and the Holst Centre, also operate as IP orchestrators, allowing multiple (private and public) parties to invest in and join large R&D programs [66]. ESI is a research center for systems design and engineering in the high-tech equipment industry, which closely collaborates with industrial companies as well as academic institutes [67]. The Holst Centre is a research center that develops generic health technologies based on flexible and wireless electronics [68].

Notably, the belief that entrepreneurship in the "new economy" primarily takes place in the digital world is widespread. Consequently, the implicit assumption often is that ecosystems for software-driven startups, such as those in Silicon Valley and Berlin, are the gold standard for entrepreneurship policy [11,69]. However, the success of the Eindhoven ecosystem suggests this widespread idea underestimates the enormous physical infrastructure required for digitalization. An example is the extensive infrastructure of sensors and fast communication systems needed for IoT, deep learning, and other forms of digitalization [70].

The exponential global development of digital products and systems [71] thus requires a complementary physical infrastructure. This also implies that the development of new deeptech systems in labs and cleanrooms is extremely location-bound, involving direct interactions between R&D professionals working on prototypes (components) of the deeptech artifact being developed. This is a direct consequence of the deeptech nature of solutions in which extremely complex hardware and software are combined. The vast majority of R&D activity in the Eindhoven ecosystem will therefore remain tied to specific locations—in line with extant theories in this area [63,64].

### 4.5. A DeepTech Identity Implies Selectivity and Coordination

The deeptech identity of Eindhoven (outlined earlier) and each of its various hotspots also implies this ecosystem has to sustain its attractiveness by being rather selective toward potential newcomers. An example is the population of companies residing at the HTCE. On the one hand, HTCE's site management team often has to say "no" to companies that would like to set up shop at the campus, but do not fit its deeptech identity [33]. On the

other hand, this management team also needs to stimulate and enable any residents that no longer fit the campus to leave. For example, in 2009 Liquavista left the HTCE because it no longer used the cleanroom facilities on the campus [34]. This selectivity in sustaining and strengthening the identity of each of the campus locations (and more broadly the Eindhoven region) is a permanent balancing act.

In addition to this selectivity, there is also a strong need for coordinating the large number of existing and new initiatives in the area of deeptech entrepreneurship. Due to the growth in campus locations, incubators, investors, and other stakeholders, some entrepreneurs knock on the wrong doors and ideas and talent may get lost. Therefore, a regional platform is currently being developed for coordination purposes. In this respect, TU/e and Fontys are currently transforming their technology transfer units into an intermediary with a portal function (called "The Gate"), centrally positioned in the regional ecosystem [72]. Moreover, TU/e has been pioneering so-called "challenge-based" forms of learning [73], also inspired by the steep learning curves observed in student-driven team projects, such as the one from which Lightyear arose [74].

*4.6. Current and Future Challenges*

The deeptech identity of the Eindhoven region (outlined earlier) and each of the various hotspots in this region also implies that it has to sustain its attractiveness by being rather selective toward potential newcomers. Overall, Eindhoven's collaborative model appears to be well equipped to address the huge technological and collaborative risks, as described in Section 2.2 (see also Figure 1). The financial risk is the most challenging one in the valley of death for deeptech entrepreneurship. Various new investment funds, like Innovation Industries and EIT InnoEnergy, have been created in the past ten years. Moreover, the Eindhoven ecosystem benefits from the presence of a substantial number of established multinational companies willing to fund and invest in deeptech ventures. For example, ASML, Philips, and other companies have been investing in HighTechXL, the deeptech venture builder located on the HTCE. Similarly, the same group of companies has recently partnered with a large Dutch pension fund and the Dutch government to create a large DeepTechXL investment fund [75]. Nevertheless, the scarcity of financial resources for bringing deeptech ventures from TRL levels 4–5 to TRL levels 7–8 remains a major challenge for the Eindhoven region as well as the larger Dutch setting [76,77].

Another major challenge in the next 5 to 10 years is the huge number of *unfilled vacancies* in multinational companies, as well as in many medium-sized and startup companies. The volume of unfilled vacancies is likely to grow further, with the total employment in the region rising with up to 70,000 additional full-time jobs over the next 10 to 15 years [78]. In this respect, the Dutch labor market for both junior as well as senior engineering talent has been out of sync for many years [79], which implies the Eindhoven region today is almost entirely dependent on foreign talent for filling the growing number of vacancies.

Moreover, various efforts to recruit foreign talent are undermined by the scarcity of *housing facilities* in the region as well as elsewhere in the Netherlands. One ongoing project is the transformation of a key part of Eindhoven's city center in order to enable a much larger number of inhabitants to live in this area [80]. In addition, the Dutch government recently committed resources to build 45,000 homes in the Eindhoven region. However, this planned growth in housing infrastructure will only partly meet the existing demand for homes and apartments. The continued high growth rate of regional employment will thus sustain the mismatch between supply and demand in the housing market for at least another ten years, and thereby constitutes an enormous challenge for spatial and urban policy makers [81].

These various problems in the basic infrastructure also raise major concerns regarding the *inclusiveness* of Eindhoven's increasing prominence and excellence as a deeptech ecosystem [78]. The growing discrepancies between demand and supply in the (local) labor and housing markets are likely to further increase the incomes of well-educated engineers and entrepreneurs as well as the average prices of houses and apartments in the

region [78], which would grow the inequalities in income and wealth and make low-income families even more vulnerable than they already are [82]. Therefore, several citizens have recently called for more democratic control by the various city councils in the region as well as deliberate policies targeting the most vulnerable citizens [83]. This call for more inclusiveness also resonates with the rise of the "quadruple helix" model, in which the triple helix model of academia, industry, and local government is extended with the fourth pillar of citizens and civil society [84,85].

## 5. Conclusions

In sum, three structural conditions appear to be critical for the emergence and success of Eindhoven's deeptech entrepreneurial ecosystem:

1.  Its focus on deeptech innovation and entrepreneurship is deeply rooted in the region's history, possessing strong competencies in systems engineering, design thinking, and multidisciplinary collaboration.
2.  The collaborative approach to regional policy, which is also based on a long cooperative tradition, gives industrial, academic and governmental actors an equivalent position in regional governance.
3.  A systemic approach toward co-locating R&D and business incubation activities in five innovation hotspots in the region.

Together, these three conditions provide a proper setting for addressing the major technological, financial and collaborative risks arising from deeptech entrepreneurship (as outlined in Section 2.2). Currently, the weakest element of Eindhoven's ecosystem is the relative scarcity of investment resources for deeptech enterprises, especially in their early stages. The lack of engineering talent and the insufficient growth of the housing infrastructure are two other major challenges.

Overall, the huge complexity of deeptech systems and products, in which very complex hardware is combined with complex software, apparently requires a truly collaborative approach at each level (e.g., firm, campus, value chain, and region) of the entrepreneurial ecosystem. Accordingly, the three structural conditions previously outlined appear to be highly complementary in supporting and enhancing the collaborative ecosystems needed for deeptech value propositions. These conditions also turn the Eindhoven case into an interesting benchmark, one that is fundamentally different from Silicon Valley, the global benchmark for technology-driven ecosystems [11–13]. As such, the Eindhoven region has the potential to provide the archetype of a collaborative ecosystem, as an alternative to the "winner-takes-all" and short-cycled entrepreneurial culture of Silicon Valley [14,15]. Future work in this area might explore whether other deeptech ecosystems arising in, for instance, France, Japan, South Korea, or Germany (e.g., [86]) are characterized by similar or distinct conditions compared to the Eindhoven case.

However, the three conditions outlined earlier also operate as boundary conditions for other regions in which the Eindhoven model might be applied. Thus, regions exclusively focusing on software-driven ventures are better off with Silicon Valley as their main benchmark. Similarly, regions that host many enterprises producing consumer products and services will also learn more from the Silicon Valley example.

Given its focus on upstream B2B markets, Eindhoven's collaborative deeptech model may therefore be less widely applicable than that of its Silicon Valley counterpart, and might merely appeal to regions that have a strong tradition in designing and developing hardware. The latter regions can infer the following insights and learnings from the Eindhoven case. Any region pursuing a deeptech profile:

(a)  needs to develop a clear regional identity in this area, one that is not swayed too much by widespread beliefs about digitalization and the "new economy";
(b)  has to create conditions for R&D and entrepreneurship which do justice to this profile, by structurally investing in knowledge-based and professional competencies that have a high time constant;

(c)　　should seek to effectively orchestrate regional governance without turning it into a core task of local government(s).

The Eindhoven case also raises major concerns about whether the most vulnerable citizens benefit from the ecosystem's growth and success. Thus, to become a truly convincing alternative for the Silicon Valley benchmark, Eindhoven needs to stretch its collaborative strengths in order to turn it into a globally leading, as well as inclusive, deeptech ecosystem.

**Funding:** This research received no external funding.

**Institutional Review Board Statement:** Not applicable.

**Data Availability Statement:** More details on the (collection and analysis of the) data used and reported in this article can be found in [22,23,34] and [35–44]. All these documents are fully open-access available via the links in the References.

**Acknowledgments:** The author acknowledges the helpful comments and feedback on an earlier draft by Myriam Cloodt and Robert-Jan Smits as well as three reviewers of this journal.

**Conflicts of Interest:** The author declares no conflict of interest.

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
