# Peer review of "Against All Odds: How Eindhoven Emerged as a Deeptech Ecosystem"

_systems, doi:10.3390/systems10040119_

Round 1
Reviewer 1 Report
I found this paper very relevant and interesting, and it can be published as it reads now, so no changes are required.
As the author points out, it is important to balance the myth of Silicon Vallay as the prototype of entrepreneurship, and thus this article is very welcome, as there are very few articles looking at hard ware/deeptech entrepreneurship. This is especially important in a Nordic/continental European context as this part of Europe does not have a platform economy based on digital software innovations but a manufacturing economy based on hardware/deeptech innovations. This article is an excellent description of this situation, and it deals with some of the important challenges, like long-term and patient funding, that deep tech entrepreneurs are exposed to.
Author Response
Many thanks for your constructive review and assessment! Inspired by your reflection and comments, I adapted the Introduction section accordingly.
Reviewer 2 Report
the paper deals with a pertinent and interesting topic, and it is well written. It has also some characteristics that need to be dealt with: it is descriptive and based on anecdotal information and previous experience.
The main problem with the paper is that it reads as an essay and not as a journal article. Moreover, its weakest link is the methods section. Why resting on 20 master thesis is good background support? Supervising, for example, only 15 is not knowledge-base enough?
15 years of participant observation is good enough? Let me put it very simply: just because I have 16 years of a totally different experience, with a totally different perspective, does not make me more knowledgeable and closer to the 'truth'/reality than the author, does it?
I am sorry, the author needs to support his point on academic grounds.
Just anecdotal information is not enough!
Part of what's being written on page 10 [457-462] is almost a copy/paste of what's on page 2 [60-66]
Conclusions are to conclude. Avoid references in the conclusion section.
Author Response
Many thanks for your review! Your comments about the (e.g. participant-observation) method used in this paper are legitimate. However, please note this manuscript was indeed submitted as an "essay" rather than a regular paper for Systems. The revised paper uses the complete layout of a Systems article, with the label Essay (as Type of Paper) in its heading on the first page, to make this clear upfront.
Please also note that (almost) all major observations reported in the manuscript are based on other studies/reports/articles. My main contribution in this essay is to synthesize this fragmented body of knowledge in a coherent argument.
To position the methods used in the paper better, I've added a short text in the Method section of the revised manuscript to clarify the role of the "participant-observation" data in the way I've collected and analyzed data. The revised section 3 thus now reads as follows:
"In this essay, I draw on three sources of data. First, earlier work includes studies of the largest and most prominent deep-tech campus in the Eindhoven region [33,34] and a study conducted for the OECD in which five entrepreneurial ecosystems (incl. Eindhoven) in The Netherlands were analyzed and compared [22,23]. The latter study draws on both quantitative and qualitative data regarding regional governance, knowledge flows, collaborative practices and other key aspects of these five ecosystems.
Second, this essay also draws on more than twenty MSc thesis projects on various innovation and entrepreneurial practices in the Eindhoven ecosystem, all (co)supervised by the author of this article. For the findings reported in the remainder of this paper, the ten most important MSc theses are [35-44]. These thesis projects employed case study methods, mathematical simulation models, surveys, and other research methods to codify and analyze various entrepreneurial practices. For more detailed information, I refer to the full thesis reports [35-44], all available in open access.
Third, this essay benefits from more than 15 years of participant-observation [45] by the author. For example, preliminary versions of the assessment described in the next section were used in many presentations and discussions with visiting delegations from other regions [46], which allowed the author to further validate the findings and conclusions about the Eindhoven ecosystem reported in this paper. Notably, the evidence collected in the first two sets of studies forms the backbone of the key argument in the next section; the participant-observation role primarily served to triangulate this key argument, by obtaining input from outsiders (e.g., with extensive knowledge of how other regional ecosystems operate)."
Reviewer 3 Report
I read the essay with great interest.
The material draws attention to the problem of sustainable development of knowledge-intensive entrepreneurship. The author notes the critical stages in the development of technology, expressed in TRL. The objective problems related to investment, human, infrastructural resources that hinder the development of entrepreneurial idea and often lead to the death of business are highlighted. This entails the inhibition of scientific and technological progress as a whole, as many significant ideas do not survive to commercialization.
The usefulness of the work is defined by the example of an advanced district in the Netherlands. This example instructively defines the rules of effective development of the territory.
The author relies on the graduate qualification works of undergraduates, publications in the open press about the history and problems of Eindhoven development, compares Eindhoven with Silicon Valley. The result is both objective and subjective conclusions about the specifics of development and improvement of the target region. This corresponds to the chosen genre of work.
At the same time, there are other points of powerful growth in the world, which may not have had their own name, but the specifics of the breakthrough development of states would also be interesting in comparison with Eindhoven. I mean the activities of Japan, South Korea, China, Taiwan, Germany, Italy, etc. I think such comparisons would further enrich the work.
At the same time, I consider it possible to publish this essay.
Good luck with further research!
Author Response
Many thanks for your helpful assessment of the manuscript!
Regarding your comment about other locations/regions "of powerful growth in the world": I agree that it would be very interesting to compare with other regions (beyond Silicon Valley). I've therefore inserted a sentence in the last section, calling for such future work; this new sentence is as follows:
"Future work in this area can explore whether other deeptech ecosystems arising in, for instance, France, Japan, South Korea, or Germany [e.g., 86] are characterized by similar or distinct conditions compared to the Eindhoven case."
Round 2
Reviewer 2 Report
Thanks for the clarification and effort.